# Tumor Cavitation with Anlotinib Treatment in Lung Adenocarcinoma

**DOI:** 10.3390/diagnostics15101280

**Published:** 2025-05-18

**Authors:** Jie Huang, Xueqin Chen

**Affiliations:** 1Department of Thoracic Oncology, Hangzhou Cancer Hospital, No. 34, Yanguan Lane, Shangcheng District, Hangzhou 310002, China; drhunterjefferson@gmail.com; 2Affiliated Hangzhou First People’s Hospital, School of Medicine, Westlake University, Hangzhou 310006, China

**Keywords:** lung adenocarcinoma, anlotinib, tumor cavitation, anti-angiogenic effects

## Abstract

Tumor cavitation is distinguished by the emergence of central necrosis and cavity formation within the tumor mass, which indicates a notable outcome of anti-angiogenic therapies. This case describes a 52-year-old Chinese female with advanced EGFR-mutated lung adenocarcinoma (Exon 19 deletion), which was metastatic to bilateral lungs, brain, and right adrenal gland, who exhibited a radiographic response to combination therapy with the third-generation EGFR tyrosine kinase inhibitor (TKI) aumolertinib and the anti-angiogenic agent anlotinib. The patient achieved near-complete cavitation of almost all bilateral lung nodules, manifesting as distinctive “bullet hole” lesions on the chest CT. Despite this initial response, disease progression occurred two months later with new liver metastases, culminating in the patient’s death. This case underscores the potential efficacy of EGFR TKIs and anti-angiogenic agents in inducing unique tumor microenvironment modifications, while highlighting the transient nature of such responses and the critical need to address resistance mechanisms. Tumor cavitation may serve as a radiographic marker of anti-angiogenic activity but does not preclude metastatic spread, necessitating vigilant monitoring even in the setting of favorable imaging changes.

A 52-year-old Chinese female patient with advanced lung adenocarcinoma harboring an EGFR Exon 19 deletion mutation presented with metastases to the bilateral lungs, brain, and right adrenal gland. She was a never-smoker with no working exposition and cancer familiarity. Initial treatment with the third-generation EGFR tyrosine kinase inhibitor (TKI) furmonertinib resulted in disease progression within 7 months.

Subsequent administration of standard first-line and second-line systemic chemotherapies also failed to arrest tumor progression. Consequently, the patient received a combination therapy comprising the orally taken third-generation EGFR TKI aumolertinib at a dose of 110 mg daily and the small-molecule angiogenesis inhibitor anlotinib [1] at a dose schedule of 12 mg daily, 2 weeks on/1 week off. Baseline chest computed tomography (CT) scans revealed multiple solid metastatic nodules in the bilateral lungs (Figure 1A). Notably, after 6 months of this combination therapy, the chest CT scans demonstrated near-complete cavitation of the lung nodules, manifesting as “bullet holes” (Figure 1B). Along with the remarkable imaging feature, which was attributed to the anti-angiogenic effects of anlotinib, the CEA levels also dropped from 650 ng/mL to 430.2 ng/mL, and the patient reported a partial relief of shortness of breath.

Angiogenesis, the process of new blood vessel formation, is a well-established characteristic of the tumor microenvironment. Anti-angiogenic agents, including bevacizumab, sorafenib, apatinib, and anlotinib, have been extensively utilized in clinical settings. These pharmacological interventions aim to disrupt the tumor-associated vasculature, thereby facilitating vascular normalization [2]. A notable outcome of anti-angiogenic therapy, especially with agents such as sorafenib, regorafenib, and apatinib, as well as anlotinib, is tumor cavitation, which is distinguished by the emergence of central necrosis and the formation of cavities within the tumor mass [3,4,5,6]. As stated in a retrospective analysis of 124 lung cancer patients who were treated with anti-angiogenesis agents in MD Anderson Center, about 14% (17) patients developed tumor cavitations [4]. Although tumor cavitation has also been reported to be induced by other treatment strategies such as immunotherapy, the frequency is still low. However, to our knowledge, such an interesting image with near-complete cavitation of almost all lung nodules was never reported.

Despite the initial promising response, the patient’s disease unfortunately progressed two months later, with the emergence of liver metastases and ultimately death. This case highlights the potential efficacy of combining anti-angiogenic agents with targeted therapies like EGFR TKIs in EGFR-mutated advanced lung cancer, as evidenced by the remarkable tumor cavitation observed with the aumolertinib and anlotinib combination. However, the eventual progression underscores the need for further research into overcoming resistance mechanisms and developing more durable therapeutic strategies. We acknowledge the limitations related to the single-case design and absence of other information, including the PD-L1 status and resistance mutation details, due to the patient’s preferences. While tumor cavitation may signify a favorable initial response to anti-angiogenic agents, it does not consistently indicate a good prognosis. Thus, intensive monitoring for metastasis is imperative, as new metastatic lesions can develop despite this radiographic finding.

## Figures and Tables

**Figure 1 diagnostics-15-01280-f001:**
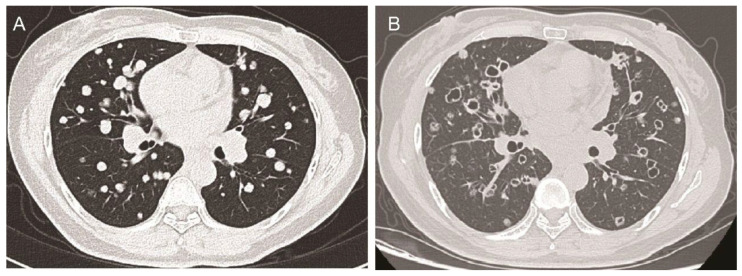
The chest CT scans of the patient before and after anlotinib treatment. (**A**): Axial chest CT scan showing multiple solid metastatic nodules in bilateral lungs at the baseline before anlotinib treatment. (**B**) After 6 months of anlotinib treatment, the axial chest CT scan shows that almost all nodules with tumor cavity looked like holes shot by bullets.

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
