# Peer review of "Tumor Cavitation with Anlotinib Treatment in Lung Adenocarcinoma"

_diagnostics, 2025, doi:10.3390/diagnostics15101280_

Round 1

Reviewer 1 Report

Comments and Suggestions for Authors

The manuscript presents a rare and clinically significant case of near-complete tumor cavitation in bilateral lung metastases following combination therapy with anlotinib and aumolertinib. While the radiographic documentation of "bullet hole" lesions is compelling, several critical revisions are required to strengthen the scientific rigor and clinical relevance of this report.

Major comments: 

  1. The introduction would benefit from a more comprehensive discussion of tumor cavitation as a biomarker of anti-angiogenic therapy. Key studies, such as Jain’s work on vascular normalization, should be cited to contextualize the observed phenomenon. Additionally, clarify how anlotinib's potent VEGFR inhibition correlates with tumor cavitation and provide comparative evidence against other anti-angiogenic agents.
  2. Methodological details require expansion. The dosage and administration schedule of anlotinib (e.g., 12 mg/day, 2 weeks on/1 week off) should be explicitly stated. The imaging evaluation criteria (e.g., RECIST 1.1) must be defined to standardize response assessment and ensure reproducibility.
  3. While the CT images demonstrate striking cavitation, quantitative data are lacking. Longitudinal measurements of lesion size before and after treatment, alongside tumor marker trends (e.g., CEA), would strengthen the correlation between imaging findings and therapeutic response.
  4. The discussion section appropriately highlights the transient nature of the response but overstates the "synergistic efficacy" of the combination therapy without direct mechanistic evidence.
  5. Limitations related to the single-case design and absence of molecular profiling (e.g., PD-L1 status, resistance mutations) should be clearly articulated. 

Minor comments: 

  1. Redundant Expressions: Issue: "Central necrosis and the formation of cavities" (Abstract line 11-12) contains overlapping descriptors. Revision: "Central necrosis and cavity formation" (Simplify phrasing).
  2. Inconsistent Capitalization: Issue: "EGFR exon 19 deletion mutation" (Abstract line 13-14) uses lowercase for "exon," while similar terms in oncology often capitalize key descriptors. Revision: "EGFR Exon 19 deletion mutation" (Capitalize genomic regions per standard conventions).
  3. Incorrect Terminology: Issue: The phrase "third-line EGFR TKI aumolertinib" (Line 16, 34) and "third-line EGFR tyrosine kinase inhibitor (TKI) furmonertinib" (Line 30) incorrectly uses "third-line" to describe the drug’s generation. Revision: Replace "third-line" with "third-generation" to accurately reflect the drug’s classification: "third-generation EGFR tyrosine kinase inhibitor (TKI) aumolertinib/furmonertinib". 
Comments on the Quality of English Language

The manuscript is generally comprehensible but requires moderate language revisions to meet academic writing standards. Key issues include: 

  1. Redundant Expressions: Issue: "Central necrosis and the formation of cavities" (Abstract line 11-12) contains overlapping descriptors. Revision: "Central necrosis and cavity formation" (Simplify phrasing).
  2. Inconsistent Capitalization: Issue: "EGFR exon 19 deletion mutation" (Abstract line 13-14) uses lowercase for "exon," while similar terms in oncology often capitalize key descriptors. Revision: "EGFR Exon 19 deletion mutation" (Capitalize genomic regions per standard conventions).
  3. Incorrect Terminology: Issue: The phrase "third-line EGFR TKI aumolertinib" (Line 16, 34) and "third-line EGFR tyrosine kinase inhibitor (TKI) furmonertinib" (Line 30) incorrectly uses "third-line" to describe the drug’s generation. Revision: Replace "third-line" with "third-generation" to accurately reflect the drug’s classification: "third-generation EGFR tyrosine kinase inhibitor (TKI) aumolertinib/furmonertinib". 

Author Response

The manuscript presents a rare and clinically significant case of near-complete tumor cavitation in bilateral lung metastases following combination therapy with anlotinib and aumolertinib. While the radiographic documentation of "bullet hole" lesions is compelling, several critical revisions are required to strengthen the scientific rigor and clinical relevance of this report.

Response: We thank the reviewer for recognition of the clinical significance and pointing out the weakness of our interesting image. We have revised our manuscript accordingly.

Major comments: 

  1. The introduction would benefit from a more comprehensive discussion of tumor cavitation as a biomarker of anti-angiogenic therapy. Key studies, such as Jain’s work on vascular normalization, should be cited to contextualize the observed phenomenon. Additionally, clarify how anlotinib's potent VEGFR inhibition correlates with tumor cavitation and provide comparative evidence against other anti-angiogenic agents.

Response: We thank the reviewer for the insightful suggestions. We have enriched the discussion of tumor cavitation as a biomarker of anti-angiogenic therapy including anlotinib and other anti-angiogenic agents, as shown in Page 2 Line 52-59. We agree that Jain’s work on vascular normalization is critical in the anti-angiogenesis field and we add the citation in Page 2 Line 52 “These pharmacological interventions aim to disrupt the tumor-associated vasculature, thereby facilitating vascular normalization2.”

  1. Methodological details require expansion. The dosage and administration schedule of anlotinib (e.g., 12 mg/day, 2 weeks on/1 week off) should be explicitly stated. The imaging evaluation criteria (e.g., RECIST 1.1) must be defined to standardize response assessment and ensure reproducibility.

Response: Thank you for your reminder. We have added the methodological details about dosage and administration schedule, as shown in Page 1 Line 33-35. “Consequently, the patient received a combination therapy comprising the orally taken third-generation EGFR TKI aumolertinib at a dose of 110mg daily and the small mole-cule angiogenesis inhibitor anlotinib1 at a dose schedule of 12 mg daily, 2 weeks on/1 week off.” Regarding to the imaging evaluation, we agree that RECIST 1.1 is commonly used as a imaging evaluation criteria of response assessment, however, tumors with cavitation are usually avoided to be chosen as a target lesion since the measurement of the lesion size is quite difficult for those lesions.

  1. While the CT images demonstrate striking cavitation, quantitative data are lacking. Longitudinal measurements of lesion size before and after treatment, alongside tumor marker trends (e.g., CEA), would strengthen the correlation between imaging findings and therapeutic response.

Response: We thank the reviewer for the suggestions. We added the CEA level changes before and after the treatment. And the patient got a relief of the shortness of breath, which indicates the therapeutic response. However, the measurement of the lesion size is quite difficult for those lesions since those lesions all developed near-complete cavitations. The tumor cavitation showed a sign of response; however, the tumor cavitation did not clearly indicate a good PFS. The patient developed liver metastasis and death very soon after the treatment although with the near-complete tumor cavitation in our study. as discussed in the discussion part in Page 2 Line 67-70, we mentioned that “while tumor cavitation may signify a favorable initial response to anti-angiogenic agents, intensive monitoring for metastasis is imperative, as the development of new metastatic lesions can occur despite this promising radiographic finding”.

  1. The discussion section appropriately highlights the transient nature of the response but overstates the "synergistic efficacy" of the combination therapy without direct mechanistic evidence.

Response: Thank you for the comments. We agree that we did not show any mechanistic evidence in this case and we have revised the descriptions, as shown in Page 2 Line 62-64.

  1. Limitations related to the single-case design and absence of molecular profiling (e.g., PD-L1 status, resistance mutations) should be clearly articulated.

Response: Thank you for the comments. We have added the limitations, as shown in Page 2 Line 67-69.

Minor comments: 

  1. Redundant Expressions: Issue: "Central necrosis and the formation of cavities" (Abstract line 11-12) contains overlapping descriptors. Revision: "Central necrosis and cavity formation" (Simplify phrasing).

Response: Thank you for the comments. We have revised the expressions, as shown in Page 1 Line 11-12.

  1. Inconsistent Capitalization: Issue: "EGFR exon 19 deletion mutation" (Abstract line 13-14) uses lowercase for "exon," while similar terms in oncology often capitalize key descriptors. Revision: "EGFR Exon 19 deletion mutation" (Capitalize genomic regions per standard conventions).

Response: Thank you for the comments. We have revised the expressions, as shown in Page 1 Line 13-14.

  1. Incorrect Terminology: Issue: The phrase "third-line EGFR TKI aumolertinib" (Line 16, 34) and "third-line EGFR tyrosine kinase inhibitor (TKI) furmonertinib" (Line 30) incorrectly uses "third-line" to describe the drug’s generation. Revision: Replace "third-line" with "third-generation" to accurately reflect the drug’s classification: "third-generation EGFR tyrosine kinase inhibitor (TKI) aumolertinib/furmonertinib". 

Response: Thank you for the comments. We have revised the expressions, as shown in Page 1 Line 16, 30, 34.

Comments on the Quality of English Language

The manuscript is generally comprehensible but requires moderate language revisions to meet academic writing standards. Key issues include: 

  1. Redundant Expressions: Issue: "Central necrosis and the formation of cavities" (Abstract line 11-12) contains overlapping descriptors. Revision: "Central necrosis and cavity formation" (Simplify phrasing).

Response: Thank you for the comments. We have revised the expressions, as shown in Page 1 Line 11-12.

  1. Inconsistent Capitalization: Issue: "EGFR exon 19 deletion mutation" (Abstract line 13-14) uses lowercase for "exon," while similar terms in oncology often capitalize key descriptors. Revision: "EGFR Exon 19 deletion mutation" (Capitalize genomic regions per standard conventions).

Response: Thank you for the comments. We have revised the expressions, as shown in Page 1 Line 13-14.

  1. Incorrect Terminology: Issue: The phrase "third-line EGFR TKI aumolertinib" (Line 16, 34) and "third-line EGFR tyrosine kinase inhibitor (TKI) furmonertinib" (Line 30) incorrectly uses "third-line" to describe the drug’s generation. Revision: Replace "third-line" with "third-generation" to accurately reflect the drug’s classification: "third-generation EGFR tyrosine kinase inhibitor (TKI) aumolertinib/furmonertinib". 

Response: Thank you for the comments. We have revised the expressions, as shown in Page 1 Line 16, 30, 34.

Reviewer 2 Report

Comments and Suggestions for Authors

The Authors describe a case report of a female patient with advanced lung adenocarcinoma presenting multiple cavitation detected by CT scan. I have minor comments:

  • please specify if the third line is according to guidelines
  • please add patient's risk of factor (smoking, working exposition, familiarity...)
  • please expand molecular tumor characterization
  • in figure 1, please use images of the same level (nodule are not perfectly corresponding in the 2 panels)
  • I think that it is usefìul to expand discussion about angiogenetic effect and use more references, in particular about lung cancer cavitation
  • There are conflicting data in the literature on the prognostic significance of cavitate lesions in lung cancer. please discuss briefly

Author Response

  • please specify if the third line is according to guidelines

Response: We thank the reviewer for the comments. According to the CSCO and CACA guideline in China, anlotinib treatment is recommended as a third-line treatment strategy after two lines of chemotherapy. 

  • please add patient's risk of factor (smoking, working exposition, familiarity...)

Response: We thank the reviewer for the comments. We have added the patient’s risk of factor, as shown in Page 1 Line 30-31.

  • please expand molecular tumor characterization

Response: We thank the reviewer for the comments. As described in Page 1 Line 29, “A 52-year-old Chinese female patient with advanced lung adenocarcinoma har-boring an EGFR exon 19 deletion mutation presented with metastases to the bilateral lungs, brain, and right adrenal gland”. We added the limitations that the PD-L1 status and molecular profiling after EGFR-TKI resistance are unavailable due to the patient’s choice, as shown in Page 2 Line 68-70.

  • in figure 1, please use images of the same level (nodule are not perfectly corresponding in the 2 panels)

Response: We thank the reviewer for the comments. We admit the nodules were not perfectly corresponding, however, the two CT images we showed were the closest transverse sections we could find.

  • I think that it is usefìul to expand discussion about angiogenetic effect and use more references, in particular about lung cancer cavitation

Response: We thank the reviewer for the insightful suggestions. We have enriched the discussion of tumor cavitation as a biomarker of anti-angiogenic therapy in lung cancer including anlotinib and other anti-angiogenic agents with more references, as shown in Page 2 Line 52-59.

  • There are conflicting data in the literature on the prognostic significance of cavitate lesions in lung cancer. please discuss briefly

Response: We thank the reviewer for the insightful suggestions. We discussed in Page 2 Line 70-74. “While tumor cavitation may signify a favorable initial response to anti-angiogenic agents, tumor cavitation does not consistently indicate a good prognosis, intensive monitoring for metastasis is imperative, as the development of new metastatic lesions can occur despite this promising radiographic finding”.

Round 2

Reviewer 1 Report

Comments and Suggestions for Authors

Revised General Remarks:
The authors have effectively addressed previous feedback by improving the discussion of mechanisms, clarifying methods, and openly acknowledging limitations. The unique "bullet-hole" imaging findings highlight the study's key strength.

Revisions Required

  1. CEA Unit Correction:

    • Issue: "650 mg/ml" → 650 ng/mL (Line 42)  (common CEA units are nanograms/mL; Please further verify to ensure academic rigor.).
    • Action: Revise all instances to ensure clinical data accuracy.
  2. Minor Formatting Error:

    • Issue"the formation of cavities within the tumor mass.[3-6]."  (Line 56) → Remove the redundant period before citations.
    • Revision"the formation of cavities within the tumor mass [3-6]."
  3. Language issues: See ' Comments on the Quality of English Language '.
Comments on the Quality of English Language

The manuscript is largely clear but requires final polishing:

  1. Avoid First-Person Singular:

    • Original"to the best of my knowledge" (Line 60) 
    • Revision"To our knowledge" or "To the best of the authors’ knowledge"
  2. Sentence Structure Fix:

    • Original"While tumor cavitation may signify [...],tumor cavitation [...], intensive monitoring [...] is imperative" (Line 72) 
    • Revision"While tumor cavitation may signify a favorable initial response to anti-angiogenic agents, it does not consistently indicate a good prognosis. Thus, intensive monitoring for metastasis is imperative, as new metastatic lesions can develop despite this radiographic finding."
  3. Self-Review Suggested

Author Response

Revisions Required

  1. CEA Unit Correction:

    • Issue: "650 mg/ml" → 650 ng/mL (Line 42)  (common CEA units are nanograms/mL; Please further verify to ensure academic rigor.).
    • Action: Revise all instances to ensure clinical data accuracy.

Response: Thank you for the comments. We have checked and corrected the unit errors.

  1. Minor Formatting Error:

    • Issue"the formation of cavities within the tumor mass.[3-6]."  (Line 56) → Remove the redundant period before citations.
    • Revision"the formation of cavities within the tumor mass [3-6]."

Response: Thank you for the comments. We have checked and corrected the formatting errors.

Language issuesSee ' Comments on the Quality of English Language '.

Comments on the Quality of English Language

The manuscript is largely clear but requires final polishing:

  1. Avoid First-Person Singular:

    • Original"to the best of my knowledge" (Line 60) 
    • Revision"To our knowledge" or "To the best of the authors’ knowledge"

Response: Thank you for the comments. We have revised the sentence.

  1. Sentence Structure Fix:

    • Original"While tumor cavitation may signify [...],tumor cavitation [...], intensive monitoring [...] is imperative" (Line 72) 
    • Revision"While tumor cavitation may signify a favorable initial response to anti-angiogenic agents, it does not consistently indicate a good prognosis. Thus, intensive monitoring for metastasis is imperative, as new metastatic lesions can develop despite this radiographic finding."

Response: Thank you for the comments. We have revised the sentence.

  1. Self-Review Suggested

Response: Thank you for the comments. We have reviewed and approved the final version.